# High Density of CD16+ Tumor-Infiltrating Immune Cells in Recurrent Ovarian Cancer Is Associated with Enhanced Responsiveness to Chemotherapy and Prolonged Overall Survival

**DOI:** 10.3390/cancers13225783

**Published:** 2021-11-18

**Authors:** Alexandros Lalos, Ornella Neri, Caner Ercan, Alexander Wilhelm, Sebastian Staubli, Alberto Posabella, Benjamin Weixler, Luigi Terracciano, Salvatore Piscuoglio, Sylvia Stadlmann, Giulio C. Spagnoli, Raoul A. Droeser, Gad Singer

**Affiliations:** 1University Center for Gastrointestinal and Liver Diseases, Clarunis, University of Basel, 4031 Basel, Switzerland; alexander.wilhelm@ucsf.edu (A.W.); sebastian.staubli@usb.ch (S.S.); Alberto.Posabella@clarunis.ch (A.P.); RaoulAndre.Droeser@usb.ch (R.A.D.); 2Institute of Pathology, University Hospital Basel, 4056 Basel, Switzerland; caner.ercan@usb.ch (C.E.); luigi.terracciano@usb.ch (L.T.); 3Department of Surgery, Charité University Hospital, Campus Benjamin Franklin, 12203 Berlin, Germany; benjamin.weixler@charite.de; 4Department of Biomedicine, University Hospital Basel, 4031 Basel, Switzerland; Salvatore.Piscuoglio@usb.ch; 5Visceral Surgery Research Laboratory, Clarunis, Department of Biomedicine, 4031 Basel, Switzerland; 6Institute of Pathology, Kantonsspital Baden AG, 5404 Baden, Switzerland; sylvia.stadlmann@ksb.ch (S.S.); gad.singer@ksb.ch (G.S.); 7Istituto CNR “Translational Pharmacology”, 00133 Rome, Italy; gcspagnoli@gmail.com

**Keywords:** ovarian cancer, CD16, tissue microarray, immunohistochemistry, prognosis, biomarker, recurrence

## Abstract

**Simple Summary:**

The late—and in most cases at an advanced stage—diagnosis of patients with ovarian cancer (OC) and the high recurrence rate make this malignant disease the most lethal among gynecological cancers. With a mortality-to-incidence ratio of 0.74, OC is a tumor with the fifth most frequent progression after esophageal cancer, liver cancer, pancreatic cancer, and brain tumors. The updated FIGO staging system is the gold standard in the clinic and includes surgical, radiologic, and pathologic elements to describe the extent of OC. This system is used to describe tumor extent, plan further therapy, and predict prognosis. However, it is consistently observed that patients with identical stages and treatments have a completely different outcome in terms of survival and recurrence. This fact indicates that this classification alone is not sufficient for the prognosis of OC in the vast majority of cases. Over the last two decades, many studies have demonstrated the critical role of the tumor microenvironment in tumorigenesis, progression, prognosis, and response to chemotherapy. In the current study, we investigate the role of CD16 expression in OC.

**Abstract:**

Background: Ovarian cancer (OC) is the most aggressive and fatal malignancy of the female reproductive system. Debulking surgery with adjuvant chemotherapy represents the standard treatment, but recurrence rates are particularly high. Over the past decades, the association between the immune system and cancer progression has been extensively investigated. However, the interaction between chemotherapy and cancer immune infiltration is still unclear. In this study, we examined the prognostic role of CD16 expression in OC, as related to the effectiveness of standard adjuvant chemotherapy treatment. Methods: We analyzed the infiltration by immune cells expressing CD16, a well-characterized natural killer (NK) and myeloid cell marker, in a tissue microarray (TMA) of 47 patient specimens of primary OCs and their matching recurrences by immunohistochemistry (IHC). We analyzed our data first in the whole cohort, then in the primary tumors, and finally in recurrences. We focused on recurrence-free survival (RFS), overall survival (OS), and chemosensitivity. Chemosensitivity was defined as RFS of more than 6 months. Results: There was no significant correlation between CD16 expression and prognosis in primary carcinomas. However, interestingly, a high density of CD16-expressing tumor-infiltrating immune cells (TICs) in recurrent carcinoma was associated with better RFS (*p* = 0.008) and OS (*p* = 0.029). Moreover, high CD16 cell density in recurrent ovarian carcinoma showed a significant association with chemosensitivity (*p* = 0.034). Univariate Cox regression analysis revealed that the high expression of CD16+ TIC in recurrent cancer biopsies is significantly associated with an increased RFS (HR = 0.49; 95% CI 0.24–0.99; *p* = 0.047) and OS (HR = 0.28; 95% CI 0.10–0.77; *p* = 0.013). However, this was not independent of known prognostic factors such as age, FIGO stage, resection status, and the number of chemotherapy cycles. Conclusions: The high density of CD16-expressing TICs in recurrent ovarian cancer is associated with a better RFS and OS, thereby suggesting a previously unsuspected interaction between standard OC chemotherapy and immune cell infiltration.

## 1. Introduction

According to the global cancer statistics, ovarian cancer (OC), with 313,959 new cases in 2020, is the ninth most commonly diagnosed malignancy in women [1] and represents the most fatal tumor of the female reproductive system [2]. Due to the nonspecific symptoms and the lack of a screening test, the detection of this malignant disease is made at an advanced stage, and most patients will develop recurrences despite initially curative therapy [3].

Epithelial, germ cell, and sex-cord-stromal OC represent the three different histological types of OC, with the epithelial type being the most common (approximately 90% of the cases) [4,5]. Because of this heterogeneity, histologic interpretation of resected tissue is challenging, and evaluation by specialized pathologists is warranted. For example, carcinomas are subdivided into high-grade serous, low-grade serous, endometrioid, clear cell, and mucinous subtypes [6]. Because of this heterogeneity, some studies suggest the additional use of electron microscopy [7]. In addition, tumor differentiation grade plays a crucial role in the prognosis and overall survival in patients with OC [8,9].

The FIGO staging system, considering radiological, surgical, and pathological features, is widely used in order to describe tumor extent and predict prognosis [10,11]. OC gold standard treatment consists of debulking surgery, including total hysterectomy, bilateral salpingo-oophorectomy as well as omentectomy, followed by adjuvant chemotherapy with carboplatin and paclitaxel [12,13,14,15]. However, we consistently find that patients with identical tumor stages and treatment display entirely different clinical outcomes, thus suggesting that currently applied prognostic tools are inadequate.

Over the past 25 years, the association between the immune system, tumor microenvironment, and prognosis of solid tumors has been thoroughly investigated [16,17,18]. While in malignancies like colorectal cancer (CRC) a variety of biomarkers influencing tumor growth, and malignant cell proliferation and migration have been clearly identified [19,20], the OC microenvironment has not been investigated in similar detail [21]. Recently, we observed that a high density of OC infiltrating cells expressing CD66b, a protein that belongs to the carcinoembryonic Ag supergene family [22], independently predicts response to chemotherapy in OC [23].

Cluster of differentiation 16 (CD16), also known as FcγRIII, is a cell surface molecule expressed by a variety of immune cells, including granulocytes, macrophages, NK, and T-cell subsets binding conserved sections of IgG and mediating antibody-dependent cellular cytotoxicity (ADCC) [24,25,26,27,28,29]. Moreover, CD16-positive cells may directly recognize poorly characterized tumor ligands, also in the absence of marker-specific IgG [30]. Importantly, tumor infiltration by CD16+ myeloid cells has been shown to be associated with improved survival in colorectal carcinoma [31]. In the case of OC, it has been shown that the injection of expanded CD56^superbright^CD16+ NK cells in patient-derived xenograft ovarian cancer murine models was shown to result in tumor size reduction and improved OS [32]. These results suggest that the high expression of CD16+ cells also plays a critical role in OC.

With this in mind, in this study, we investigated the prognostic significance of OC infiltration by CD16+ cells with a particular emphasis on response to chemotherapy in primary and recurrent disease. The secondary aim of the study was to thoroughly explore the interaction between the different particles of the immune microenvironment in OC.

## 2. Materials and Methods

### 2.1. Patients

In our study, we included 47 patients with unselected, clinically annotated primary tumors and their matched recurrences. With the intention of generating a homogenous group, we included only high-grade [33] serous ovarian carcinomas (2.1% FIGO stage II, 80.9% FIGO stage III, and 17% FIGO stage IV). Treatment consisted of primary debulking surgery followed by at least three cycles of platinum-based adjuvant chemotherapy. Since recurrence occurred in all patients, we divided them into two groups according to the free interval after completion of chemotherapy. Tumors recurring less than six months after completion of adjuvant treatment were considered chemoresistant, whereas those recurring more than 6 months after completion of the chemotherapy were defined as chemosensitive [34]. Individual clinicopathological and survival data were obtained from the medical records and Gynecologic Tumor Registry.

### 2.2. Tissue Microarray Construction

The harvested tissues, which came from four different institutes in Switzerland (Institute of Pathology of the University Hospital of Basel and the Cantonal Hospitals of St. Gallen, Baden, and Liestal), were processed to construct a tissue microarray (TMA) from unselected, nonconsecutive, formalin-fixed, paraffin-embedded primary OC tissue blocks, as previously described [35]. Briefly, tissue cylinders with a diameter of 1 mm were punched from morphologically representative areas of each donor block and brought into one recipient paraffin block (30 × 25 mm) [36]. We took each punch from the center of the tumor in an area without necrosis so that each TMA spot consisted of more than 50% tumor cells.

### 2.3. Immunohistochemistry

We used standard indirect immunoperoxidase procedures (IHC; ABC-Elite, Vector Laboratories, Burlingame, CA, USA) as described in previous studies by our team [37]. Immunohistochemical (IHC) staining was performed on 4 mm sections of formalin-fixed paraffin-embedded (FFPE) recipient TMA blocks by using CD16-specific polyclonal antibodies (PA5-80622, Invitrogen, Waltham, MA, USA). Briefly, sections were pre-treated with CC1 (Ventana Medical Systems, Tucson, AZ, USA) for 16 min and incubated with a primary anti CD16 antibody at a 1:400 dilution for 20 min. The staining procedure was performed on a benchmark immunohistochemistry staining system (Ventana Medical Systems) using iVIEW-DAB as the chromogen. Areas with necrosis, artifacts, or  ≤25% of preserved tumor tissue were excluded from the analysis. Cut-off scores for low- or high-density subgroups (Figure 1) were defined by using the median value (=16 positive cells/punch).

### 2.4. Evaluation of Immunohistochemistry

Two trained research fellows (A.L. and O.N.) performed immunohistochemical analysis, and two experienced pathologists (L.T. and E.C.) validated the data independently. All of them were blinded to clinical, histopathological, and survival data. TICs were counted for each punch (approximately one high power (20×) field).

### 2.5. Statistical Analysis

Data were analyzed by using STATA software version 13 (StataCorp, College Station, TX, USA) and the Statistical Package Software R (version 4.0.2, http//.r-project.org, accessed on 24 May 2021). Cut-off values used to classify OC with low or high immune cell infiltration were available from previous publications or generated by applying regression tree analysis [38]. In this case, cut-off scores used to classify ovarian carcinomas with low or high CD16 expression were defined according to staining intensity (0 and 1 versus 2 and 3) or cut-off = 16 cells. Kruskal–Wallis, Chi-Square, or Fisher’s exact tests were used to explore the association of the clinicopathological features with the corresponding groups of the biomarker. Kaplan–Meier survival curves were compared accordingly to the log-rank test. The *p*-value was adjusted for multiple comparisons according to Benjamini and Hochberg (1995). The identification of independent predictors of RFS was tested with univariate and multivariate hazard Cox regression analyses, considering the dichotomized CD16 density. *p*-values  <  0.05 were considered statistically significant.

## 3. Results

### 3.1. Patient and Tumor Characteristics

Table 1 summarizes the clinicopathological characteristics of patients included in our study. In particular, the median age was 58 years (range: 34–77 years). Of the patients studies, 1 patient had FIGO stage II (2.1%), 38 patients had FIGO stage III (80.9%), and 8 patients had FIGO stage IV (17%) disease. Furthermore, 16 patients (34.0%) were tumor free after the debulking operation, while 17 patients (36.2%) had residual disease smaller than 2 cm and 13 patients (27.7%) had residual disease larger than 2 cm. In one patient, the residual disease status was unclear. Postoperatively, all 47 patients received at least three cycles of chemotherapy, with 39 of them (83%) receiving six or more cycles. Tumors from 33 patients (70.2%) could be classified as chemosensitive (see above), whereas those from 14 patients (29.8%) were chemoresistant. The 6-month RFS rate was 0.53 (0.38–0.66) and the 3-year OS rate was 0.47 (0.29–0.63).

### 3.2. Association of Clinicopathological Features with CD16 Expression in Primary OC

We observed a high expression of CD16 in TICs in from primary OC of 21 patients and a low expression in TICs from cancers of 22 patients with primary OC. There was no significant difference in OS and RFS between the two groups (Table 2A). Moreover, the chemosensitivity of primary tumors was not correlated with high CD16 expression. All other clinicopathological features (age, FIGO stage, residual disease, numbers of chemotherapy cycles, and response to chemotherapy) were similarly distributed in the two groups.

### 3.3. Association of Clinicopathological Features with CD16-Positive TIC Density in Recurrent OC

To further investigate the role of CD16 expression in OC, we compared the patients with high and low CD16 density in the matched recurrent carcinomas. General clinicopathological features (age and FIGO stage) were equally distributed between the two groups. However, we found that patients with high expression of CD16 in TICs in their recurrent biopsies had a significantly longer OS compared to the patients with low CD16 expression (52.8 vs. 29.0 months, *p* = 0.008). Moreover, high CD16 cell density in recurrent ovarian carcinoma showed a significant association with chemosensitivity (*p* = 0.034). Details of CD16 expression in association with clinicopathological data are illustrated in Table 2B for recurrent cancer biopsies.

### 3.4. Correlation of CD16 TIC with Other Cells of the Immune Microenvironment

To further elucidate the immune microenvironment in high CD16 biopsies, we performed a Spearman correlation analysis with other immune markers [22,35], but no significant correlation was detectable in primary cancer biopsies. In sharp contrast, in recurrent cancer biopsies, there was a significant correlation with the expression of CD66b, IL-17, MPO, and CXCR4 (Table 3A,B).

### 3.5. Prognostic Significance of CD16 Expression in Recurrent OC

Kaplan–Meier plots clearly indicated that the recurrence-free and overall survival were significantly improved in recurrent OC with a high density of CD16 TICs, compared to tumors showing a low density (HR = 0.28; 95% CI 0.28–0.77; *p* = 0.013) (Figure 2). However, in a multivariate Hazard Cox regression survival analysis, CD16 expression could not retain its role as an independent prognostic factor for OS. FIGO stages and age were devoid of prognostic significance in univariate and multivariate analyses (Table 4).

## 4. Discussion

Despite the combination of debulking surgery and adjuvant chemotherapy, OC is still characterized by poor prognosis [39] and requires an intensive follow-up [40]. Tumor recurrences are frequently treated by chemotherapy [41,42], but responsiveness is highly heterogeneous [43,44,45]. In this context, it is important to point out the high toxicity of these therapies, which has a devastating impact on patients’ quality of life, especially when this treatment is ultimately ineffective. Therefore, the identification of markers predicting the response of recurrent tumors is urgently required.

Although we have made significant progress in understanding the role of the microenvironment and immune cells in the progression of malignancies over the past few decades, there are still controversial issues that warrant further investigation. Tumor-associated macrophages (TAMs) and tumor-associated neutrophils (TANs) have been implicated in both promoting and inhibiting tumor growth [46]. Lehman et al. showed that the tissue environment determines which cellular effector pathways are responsible for antibody-dependent tumor immunotherapy. Their results suggest that TAMs may play a dual role: not only do they promote tumor growth in certain tissue environments, but they also contribute to tumor cell destruction during antibody-mediated immunotherapy.

In our study, we investigated the expression of CD16 in tumor-infiltrating cells (TICs), aiming at identifying markers predicting the response to therapy in primary and recurrent ovarian cancer.

CD16 (Fc gamma RIII) is highly expressed by NK cells and to moderate levels by granulocytes, tissue macrophages, and subsets of monocytes, eosinophils, and dendritic cells [26]. CD16 mediates antibody-dependent cellular cytotoxicity (ADCC) and antibody-dependent phagocytosis (ADP) by NK cells and macrophages, respectively. Moreover, it has been demonstrated to directly recognize poorly characterized tumor ligands [47]. CD16 is reported to be the most potent activating receptor on freshly isolated human NK cells, able to elicit strong cytotoxicity and cytokine production [24]. Indeed, although the role of CD16-mediated ADCC may differ among cancer stages, a correlation between CD16 polymorphism and the clinical efficacy of therapeutic antibodies has been reported [48,49]. However, infiltration by NK cells in primary OC is limited [50].

The interactions between the different particles of the microenvironment are crucial. In the case of NK cells, it has been shown that their full activation requires the interaction of different cell-surface receptors [28]. Interestingly, in the recurrent cancer biopsies of our cohort, there was a significant correlation with CD66b, IL-17, MPO, and CXCR4, suggesting a stronger immune response in samples with high CD16. With this study, we aim to highlight the importance of the immune microenvironment in cancer development and demonstrate the correlation and predictive significance of a high density of CD16-positive tumor-infiltrating immune cells with RFS and OS. To date, numerous studies have shown that the immune response is a favorable prognostic factor for the clinical outcome of OC [51]. The goal would be to combine reliable biomarkers, which predict the response to treatment and lead to individualized therapy [52]. Henriksen et al. also investigated the role of NK cells expressing CD16 at high levels, but this time in peripheral samples, and found that a low blood NK cell counts were associated with an unfavorable prognosis in recurrent metastatic ovarian cancer during chemotherapy [53]. Similarly, an increase in CD16+ populations in the peripheral blood of patients with lymphoma, breast cancer, and colorectal cancer has been associated with a better prognosis [54,55,56]. Although this is a different approach, it correlates with our findings on the relationship between immune response and chemo-responsiveness.

Our data show for the first time that a high density of CD16-positive TICs represents a favorable prognostic marker in recurrent ovarian cancer and an indirect marker of tumor chemosensitivity. Intriguingly, CD16+ immune cell infiltration was devoid of prognostic relevance in primary OC. We might therefore speculate that chemotherapy is able to induce, in a subset of patients, the recruitment and/or the activation of CD16+ immune cells, influencing in turn the effects of subsequent chemotherapy cycles.

The functional significance of myeloid cells, including high CD16 expressors, is hotly debated. Lung cancer infiltration is associated with poor prognosis [57], whereas infiltration of colorectal cancer by myeloid cells correlates with a favorable clinical course [31,58,59]. Our data contribute to this debate by identifying an additional variable, represented by earlier chemotherapy treatment, of prognostic relevance in recurrent cancers.

Our study has several limitations. First, it is a retrospective study. However, emerging data might help to develop prospective studies. Second, the sample size is small, and therefore our results need to be validated in larger patient cohort to add more power to our results and facilitate their generalization. Indeed, using the data resulting from this comprehensive retrospective analysis, we plan to study a new independent cohort of patients to validate our results, which will be part of an ongoing prospective study. Finally, there was a significant difference in the residual disease between the two groups, which could explain our finding. However, univariate Hazard Cox regression indicates that residual disease status does not significantly affect the OS in our cohort. Nevertheless, these findings might pave the way towards innovative studies, addressing the prognostic and predictive significance of immune cell infiltration within the context of tumor chemotherapy.

## 5. Conclusions

In conclusion, we demonstrate with this study that the high density of CD16-expressing TICs in recurrent ovarian cancer samples is associated with significantly better RFS and OS, suggesting a previously unsuspected interaction between standard OC chemotherapy and immune cell infiltration. Our results might be the rationale for using CD16-expressing TICs as a prognostic marker for recurrent ovarian cancer and as an indirect marker for tumor chemosensitivity. Furthermore, these results could be used for the development of novel treatment modalities by modifying the tumor immune microenvironment in OC patients, especially in the context of personalized medicine.

## Figures and Tables

**Figure 1 cancers-13-05783-f001:**
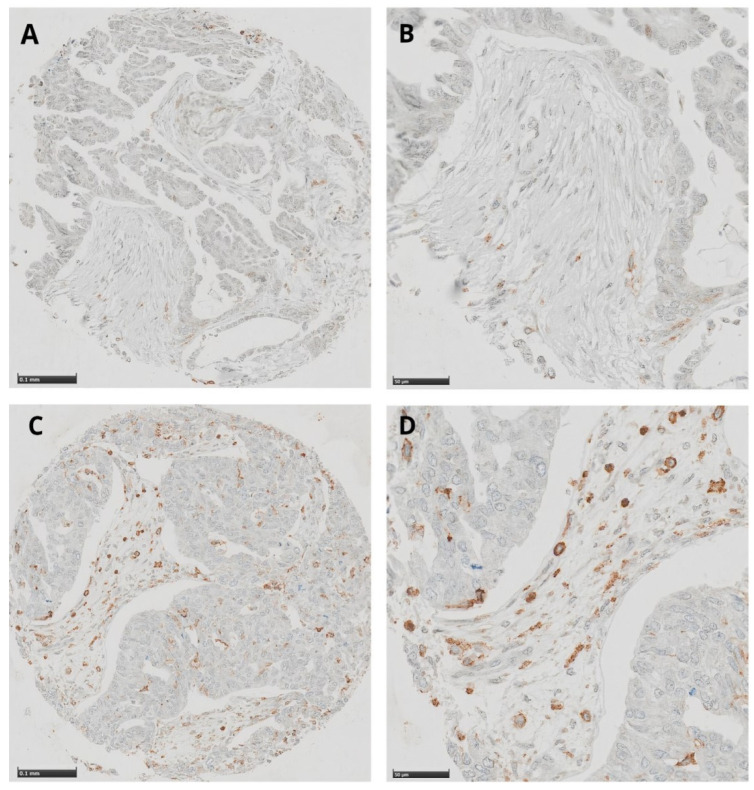
(**A**,**B**) Images demonstrate a high CD16 expression in TICs, whereas (**C**,**D**) images show only low to moderate CD16 expression in TICs. TICs: tumor-infiltrating cells.

**Figure 2 cancers-13-05783-f002:**
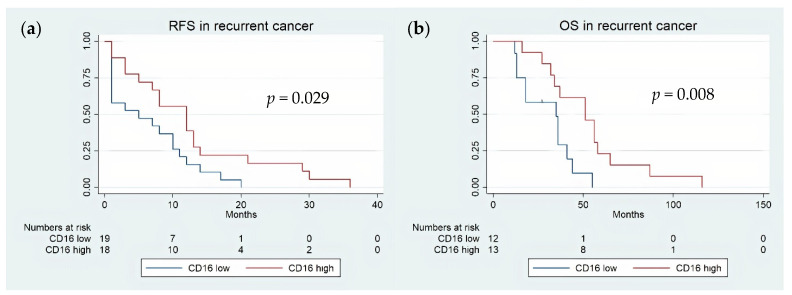
(**a**) Kaplan–Meier survival curve of recurrence-free survival according to CD16 tumor-infiltrating immune cell density in recurrent ovarian cancer biopsies. (**b**) Kaplan–Meier survival curve of overall survival according to CD16 tumor-infiltrating immune cell density in recurrent ovarian cancer biopsies. Red line indicates tumors with high CD16 TIC density, whereas blue line refers to tumors with low CD16 TIC density.

**Table 1 cancers-13-05783-t001:** Patients’ characteristics of overall cohort (*n* = 47).

Characteristics	*n* = 47 (%)
Age (media, range)	58 (34–77)
FIGO stage:	1 (2.1)
II	1 (2.1)
IIIA	5 (10.6)
IIIB	32 (68.2)
IIIC	8 (17.0)
IV	
Residual disease	
None	16 (34.0)
<2 cm	17 (36.2)
≥2 cm	13 (27.7)
Unclear	1 (2.1)
Number of chemotherapy cycles	
<6	8 (17.0)
6 or more	39 (83.0)
Response to chemotherapy	33 (70.2)
CS	14 (29.8)
CR	
RFS in months	10.1 (9.89–10.30)
OS in months	41.4 (40.77–42.03)
CD16 TIC P	17.01 (15.38–18.64)
CD16 TIC R	43.16 (30.21–56.11)
CD16 Score P	105.3 (93.78–116.82)
CD16 Score R	97.2 (75.72–108.68)

Missing clinicopathological information was assumed to be missing at random. CS: chemosensitive, CR: chemoresistant, TIC: tumor-infiltrating cells, RFS: recurrence-free survival, OS: overall survival, CI: confidence interval.

**Table 2 cancers-13-05783-t002:** (**A**) Patients’ characteristics according to dichotomized distribution of CD16-positive TICs in primary cancer biopsies (cut-off = 16 cells/punch, 50th percentile, *n* = 43 *). (**B**) Patients’ characteristics according to dichotomized distribution of CD16-positive TICs in recurrent cancer biopsies (cut-off = 16 cells/punch, 50th percentile, *n* = 37 *).

**Characteristics**	**CD16^high^, *n* = 21 (%)**	**CD16^low^, *n* = 22 (%)**	***p*-Value**
Age (media, range)	58.6 (45–73)	56.4 (34–77)	0.635
FIGO stage:			
II	0 (0.0)	1 (4.5)	
IIIA	0 (0.0)	0 (0.0)	
IIIB	4 (19.0)	1 (4.5)	
IIIC	13 (61.9)	17 (77.4)	
IV	4 (19.0)	3 (13.6)	0.324
Residual disease			0.795
None	7 (33.3)	6 (27.3)	
<2 cm	8 (38.1)	8 (36.4)	
≥2 cm	5 (23.8)	8 (36.4)	
Unclear	1 (4.8)	0 (0.0)	
Number of CT cycles			0.499
<6	2 (9.5)	4 (18.2)	
6 or more	18 (85.7)	18 (81.8)	
Response to chemotherapy			
CS	15 (71.4)	15 (68.2)	0.817
CR	6 (28.6)	7 (31.8)	
Recurrence-free survival in months	10.81 (8.63–12.99)	8.36 (6.58–12.14)	0.43
Overall survival in months	36.92 (31.11–41.72)	47.87 (42.05–53.69)	0.174
(**A**)			
**Characteristics**	**CD16^high^, *n* = 18 (%)**	**CD16^low^, *n* = 19 (%)**	***p*-Value**
Age (media, range)	58.6 (45–73)	56.4 (34–77)	0.097
FIGO stage:			
II	1 (5.6)	0 (0.0)	
IIIA	1 (5.6)	0 (0.0)	
IIIB	3 (16.6)	2 (10.5)	
IIIC	9 (50.0)	14 (73.7)	
IV	4 (22.2)	3 (15.8)	0.619
Residual disease			0.017
None	6 (33.3)	9 (47.4)	
<2 cm	11 (61.1)	3 (15.8)	
≥2 cm	1 (5.6)	6 (31.6)	
Unclear	0 (0.0)	1 (5.2)	
Number of CT cycles			0.63
<6	2 (11.1)	3 (15.8)	
6 or more	16 (88.9)	15 (78.9)	
Response to chemotherapy			0.034
CS	16 (88.9)	11 (57.9)	
CR	2 (11.1)	8 (42.1)	
Recurrence-free survival in months	12.67 (10.26–15.08)	6.58 (5.17–7.99)	0.029
Overall survival in months	52.77 (46.53–59.01)	29.00 (25.74–32.26)	0.008
(**B**)			

* percentages may not add up to 100% due to missing values of defined variables; missing clinicopathological information was assumed to be missing at random. Variables are indicated as absolute numbers, %, median, or range. Age, RFS, and OS were evaluated using the Kruskal–Wallis test. FIGO stage, residual disease, numbers of chemotherapy cycles, and chemoresistance were analyzed using the Chi-square or the Fisher’s exact test. CT: chemotherapy, CS: chemosensitive, CR: chemoresistant.

**Table 3 cancers-13-05783-t003:** (**A**) Correlation analysis of CD16 with CD66b-, IL-17-, MPO-, FOXP3-, and CXCR4-positive tumor cell infiltration in primary ovarian cancer biopsies. (**B**) Correlation analysis of CD16 with CD66b-, IL-17-, MPO-, FOXP3-, and CXCR4-positive tumor cell infiltration in recurrent ovarian cancer biopsies.

**Immune Marker**		**CD66b**	**IL-17**	**MPO**	**FOXP3**	**CXCR4**
**CD16**	**rho**	0.026	0.147	0.171	0.171	0.029
	***p*-value**	0.868	0.347	0.274	0.279	0.856
(**A**)						
**CD16**	**rho**	0.395	0.328	0.331	0.353	0.441
	***p*-value**	0.016	0.048	0.045	0.060	0.008
(**B**)						

**Table 4 cancers-13-05783-t004:** Uni- and multivariate Hazard Cox regression analysis of OS in recurrent ovarian cancer.

	Univariate			Multivariate		
	HR	95% CI	*p*-Value	HR	95% CI	*p*-Value
Age	1.02	0.99–1.05	0.194	1.03	0.94–1.11	0.527
CD16 high vs. low	0.29	0.11–0.77	0.013	0.79	0.08–7.64	0.845
Residual disease < 2 cm	0.61	0.25–1.49	0.277	0.29	0.44–2.01	0.213
Residual disease ≥ 2 cm	1.28	0.49–3.38	0.616	3.54	0.66–18.94	0.139
Number of chemotherapy cycles	1.04	0.83–1.30	0.750	0.88	0.57–1.35	0.552
FIGO stage:						
IIIA	1.68	0.10–27.67	0.717	1.39	0.05–38.55	0.847
IIIB	1.43	0.15–13.99	0.759	3.17	0.25–39.89	0.371
IIIC	2.08	0.27–15.92	0.482	5.21	0.27–98.84	0.272
IV	2.03	0.23–17.59	0.52	6.97	0.44–111.16	0.17

Multivariate analyses showing Hazard ratios and *p*-value for all recurrent cancer biopsies conferred by categorized CD16 density, age, residual disease after cytoreductive surgery, number of chemotherapy cycles, and FIGO classification.

## Data Availability

The datasets used and/or analyzed during the current study are available from the corresponding author on reasonable request.

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
