# Peer review of "High Density of CD16+ Tumor-Infiltrating Immune Cells in Recurrent Ovarian Cancer Is Associated with Enhanced Responsiveness to Chemotherapy and Prolonged Overall Survival"

_cancers, 2021, doi:10.3390/cancers13225783_

Round 1
Reviewer 1 Report
The authors in this study have evaluated the prognostic significance of Ovarian Carcinoma patients' tumors infiltration by CD16+ cells with a special focus on response to chemotherapy at primary and recurrent disease. Also, they explored the interaction between the different particles of the immune microenvironment in Ovarian Carcinoma. The authors concluded that CD16 expression in OC was related to the effectiveness of standard adjuvant chemotherapy treatment. Furthermore, these results could be used for the development of novel treatment modalities by modifying the tumor immune microenvironment in OC patients.
The authors have designed and conducted the study properly and they discussed the limitations of this work; however, these limitations are considered major drawbacks in the study. Therefore, I suggested to address the following comments:
1- Simple summary is too general, I suggest to make it more related to this paper not to the other projects.
2- (TICs) some times referred as "Tumor-infiltrating immune cells " and sometimes as "tumor-initiating cells." Please adjust accordingly.
3- the sample size is small and larger samples could provide generalized finding about the study.
4- Figure 2 should be improved and high quality figures is needed.
5- The study include only only high-grade serous ovarian carcinomas (2.1% FIGO stage II, 80.9% FIGO stage III, and 17% FIGO stage IV). Why there was no including for other types.
6- Could CD16 serve as a predictive marker for primary and recurrent-ovarian carcinoma?
7- Based on literature " A low blood NK cell count in recurrent metastatic ovarian cancer during chemotherapy is associated with unfavorable prognostic impact. " How could that be compared with your work.
References" https://doi.org/10.1080/0284186X.2020.1791358"
8- An increase in the proportion of CD16-positive monocytes has been described in various infectious and inflammatory diseases
9- findings are consistent with previous studies showing an increase in CD16-positive populations has been found in the peripheral blood of patients with breast cancer, gastric cancer, cholangiocarcinoma, multiple myeloma, melanoma and chronic lymphocytic leukemia. Could you discuss more about that in the discussion part and try to correlate similar findings with your results.
10- Conclusions should be done again to give the most important findings of this work. The one which is there is too general.
Author Response
Reviewer 1
The authors in this study have evaluated the prognostic significance of Ovarian Carcinoma patients' tumors infiltration by CD16+ cells with a special focus on response to chemotherapy at primary and recurrent disease. Also, they explored the interaction between the different particles of the immune microenvironment in Ovarian Carcinoma. The authors concluded that CD16 expression in OC was related to the effectiveness of standard adjuvant chemotherapy treatment. Furthermore, these results could be used for the development of novel treatment modalities by modifying the tumor immune microenvironment in OC patients.
The authors have designed and conducted the study properly and they discussed the limitations of this work; however, these limitations are considered major drawbacks in the study. Therefore, I suggested to address the following comments:
1- Simple summary is too general; I suggest to make it more related to this paper not to the other projects.
1- We completely agree with the reviewer’s comment. We have modified the simple summary accordingly (highlighted in yellow) and made it more relevant/specific to what we are presenting in our project.
2- (TICs) sometimes referred as "Tumor-infiltrating immune cells " and sometimes as "tumor-initiating cells." Please adjust accordingly.
2- The reviewer is absolutely right with this observation. We have to thank the reviewer very much for noticing such a typo, which we overlooked. We have corrected it accordingly.
3- the sample size is small and larger samples could provide generalized finding about the study.
3- We fully agree with the reviewer’s comment about the sample size and therefore mentioned it in our limitations. A larger sample could certainly add more power to our study and facilitate generalization of our findings. Nonetheless, we plan to use the data resulting from this large retrospective analysisto study a new independent cohort of patients to validate our findings, which is part of an ongoing prospective study. For this reason, we have adjusted the discussion accordingly (highlighted in yellow).
4- Figure 2 should be improved and high quality figures is needed.
4- We thank the reviewer for his valuable contribution. Therefore, we have created new figures with a much higher quality (2574x1872px) and included them in the manuscript, replacing the old ones.
5- The study include only high-grade serous ovarian carcinomas (2.1% FIGO stage II, 80.9% FIGO stage III, and 17% FIGO stage IV). Why there was no including for other types.
5- Because high-grade serous ovarian carcinoma is the most common subtype of ovarian carcinoma (OC), we decided to analyze this subtype to obtain the most homogenous cohort. Moreover, the majority of the studies investigating the tumor microenvironment of OC are based on cohorts of high-grade serous OC specimens [Ref.: Miller et al., Nature Genetics 2020, DOI: https://doi.org/10.1038/s41588-020-0630-5].
6- Could CD16 serve as a predictive marker for primary and recurrent-ovarian carcinoma?
6- We fully agree with the reviewer’s comment and have adjusted the manuscript to make clearer the predictive role of CD16 in chemo-responsiveness of OC (highlighted in yellow in discussion and conclusion).
7- Based on literature " A low blood NK cell count in recurrent metastatic ovarian cancer during chemotherapy is associated with unfavorable prognostic impact. " How could that be compared with your work. References" https://doi.org/10.1080/0284186X.2020.1791358"
7- We thank the Reviewer for this helpful input. Indeed, there are studies that have shown an association between low blood NK cell count in recurrent metastatic disease and prognosis in OC. This result is consistent with our findings, as CD16 (Fc gamma RIII) is highly expressed by NK cells. However, we must emphasize here that in our study we investigated the expression of the CD16+ cell in tumor tissue and not in peripheral blood. As we find this paper very interesting and important, we have included it in the discussion and references (highlighted in yellow).
8- An increase in the proportion of CD16-positive monocytes has been described in various infectious and inflammatory diseases
8- We fully agree with the reviewer’s comment and consider CD16-positive cells infiltrating the tumor as a surrogate marker for the tumor immune response in the tumor microenvironment. Therefore, we hypothesize that a strong immune response in the tumor microenvironment is associated with a better prognosis in the recurrent cancer cohort due to tumor cell destruction of. At this point, we must emphasize that the patients in our cohort did not have pelvic inflammatory disease, which could influence the evaluation of histopathological samples.
9- findings are consistent with previous studies showing an increase in CD16-positive populations has been found in the peripheral blood of patients with breast cancer, gastric cancer, cholangiocarcinoma, multiple myeloma, melanoma and chronic lymphocytic leukemia. Could you discuss more about that in the discussion part and try to correlate similar findings with your results.
9- We fully agree with the reviewer’s comment and have added a paragraph on this topic to the discussion (highlighted in yellow).
10- Conclusions should be done again to give the most important findings of this work. The one which is there is too general.
10- This is correct, and therefore we have modified the conclusions accordingly (highlighted in yellow) and made them more relevant/specific to what we are presenting in our study.
Comment of the authors:
Dear Editor,
Thank you for reviewing our paper entitled: “High Density of CD16+ Tumor Infiltrating Immune Cells in Recurrent Ovarian Cancer is Associated with Enhanced Responsiveness to Chemotherapy and Prolonged Overall Survival”, which was recently submitted for publication in MDPI cancers.
Enclosed is a revised version of the manuscript that has been modified to meet the comments of the reviewers.
Hoping that the manuscript as modified meets the requirements of the reviewers and of the editor, I look forward to hearing from you.
Yours sincerely,
Alexandros Lalos

Reviewer 2 Report
The aim of this study is to analyze the progonostic significance of ovarian cancer infiltration by C16+ cells, specifically in response to chemotherapy. An other significant aim of this study is to evaluate the interaction between the different particles of the immune microenvironment in ovarian cancer.
Even if the manuscript provides an organic overview, with a densely organized structure and based on well-synthetized data, there are aspects to be mentioned, to make the article fully readable. For these reasons, the manuscript requires major changes.
Please find below an enumerated list of comments on my review of the manuscript:
LINE 67: This histological differentiation is reported by several and recent studies (see, for reference: Clinical electron microscopy in the study of human ovarian tissues – 2019). For this reason, the manuscript will benefit from providing a complete description of these subtypes of ovarian cancers, as analyzed in several morphological and ultrastructural studies.
LINE 89: Besides, experimental studies, conducted on murine models, highlighted also CD16 citotoxic potential and antitumor functions, against ovarian cancer cells. To this aim, it will be useful to discuss also this issue (see, for reference: Expanded CD56superbrightCD16+ NK cells from ovarian cancer patients are cytotoxic against autologous tumor in a patient-derived xenograft murine model – 2018).
MATERIAL AND METHODS:
As regards this section, the methodology design was rigorous and appropriately implemented within the study.
DISCUSSION:
Also this section is well organized and densely presented, based on well-synthetized data.
In conclusion, this manuscript is densely presented and well organized, based on well-synthetized data. The authors were lucid in their style of writing, making it easy to read and understand the message, portrayed in the manuscript. However, many of the topics are very concisely covered. Moreover, this research have futuristic importance and could be potential for future research. However, I have major comment only for the introductive section, for improvement before acceptance for publication. I would accept the manuscript, if the comments are addressed properly.
Author Response
Reviewer 2
The aim of this study is to analyze the prognostic significance of ovarian cancer infiltration by C16+ cells, specifically in response to chemotherapy. Another significant aim of this study is to evaluate the interaction between the different particles of the immune microenvironment in ovarian cancer.
Even if the manuscript provides an organic overview, with a densely organized structure and based on well-synthetized data, there are aspects to be mentioned, to make the article fully readable. For these reasons, the manuscript requires major changes.
Please find below an enumerated list of comments on my review of the manuscript:
LINE 67: This histological differentiation is reported by several and recent studies (see, for reference: Clinical electron microscopy in the study of human ovarian tissues – 2019). For this reason, the manuscript will benefit from providing a complete description of these subtypes of ovarian cancers, as analyzed in several morphological and ultrastructural studies.
ANSWER: We agree with the reviewer’s comment that the histopathology of ovarian carcinoma is characterized by a high degree of heterogeneity. We have described the subtypes of ovarian carcinoma in more detail, including using the reference that was suggested. The introduction has been changed accordingly (highlighted in yellow).
LINE 89: Besides, experimental studies, conducted on murine models, highlighted also CD16 cytotoxic potential and antitumor functions, against ovarian cancer cells. To this aim, it will be useful to discuss also this issue (see, for reference: Expanded CD56superbrightCD16+ NK cells from ovarian cancer patients are cytotoxic against autologous tumor in a patient-derived xenograft murine model – 2018).
ANSWER: We have to thank the reviewer for this valuable input and the reference to this experimental study, which we could use to better explain our hypothesis. The introduction has been modified accordingly (highlighted in yellow).
MATERIAL AND METHODS:
As regards this section, the methodology design was rigorous and appropriately implemented within the study.
DISCUSSION:
Also this section is well organized and densely presented, based on well-synthetized data.
In conclusion, this manuscript is densely presented and well organized, based on well-synthetized data. The authors were lucid in their style of writing, making it easy to read and understand the message, portrayed in the manuscript. However, many of the topics are very concisely covered. Moreover, this research has futuristic importance and could be potential for future research. However, I have major comment only for the introductive section, for improvement before acceptance for publication. I would accept the manuscript if the comments are addressed properly.
Comment of the authors:
Dear Editor,
Thank you for reviewing our paper entitled: “High Density of CD16+ Tumor Infiltrating Immune Cells in Recurrent Ovarian Cancer is Associated with Enhanced Responsiveness to Chemotherapy and Prolonged Overall Survival”, which was recently submitted for publication in MDPI cancers.
Enclosed is a revised version of the manuscript that has been modified to meet the comments of the reviewers.
Hoping that the manuscript as modified meets the requirements of the reviewers and of the editor, I look forward to hearing from you.
Yours sincerely,
Alexandros Lalos

Round 2
Reviewer 1 Report
Dear Authors,
You have addressed all my comments properly.
I have no further comments.
Best Regards
Reviewer 2 Report
The authors have improved the manuscript by more completely describing many of the connections they include. The organization of the manuscript is better, and the review is now more organic.